# Mitochondrial Dysfunction: The Hidden Player in the Pathogenesis of Atherosclerosis?

**DOI:** 10.3390/ijms24021086

**Published:** 2023-01-06

**Authors:** Giovanni Ciccarelli, Stefano Conte, Giovanni Cimmino, Patrizia Maiorano, Andrea Morrione, Antonio Giordano

**Affiliations:** 1Vanvitelli Cardiology Unit, Monaldi Hospital, 80131 Naples, Italy; 2Sbarro Institute for Cancer Research and Molecular Medicine, Center of Biotechnology, College of Science and Technology, Temple University, Philadelphia, PA 19122, USA; 3Department of Translational Medical Sciences, Section of Lung Diseases, University of Campania “Luigi Vanvitelli”, 80131 Naples, Italy; 4Department of Translational Medical Sciences, Section of Cardiology, University of Campania “Luigi Vanvitelli”, 80131 Naples, Italy; 5Department of Medical Biotechnologies, University of Siena, 53100 Siena, Italy

**Keywords:** mitochondrial disorders, atherosclerosis, endothelium, cardiovascular disease, coronary, aging, therapies to counteract mitochondrial dysfunction

## Abstract

Atherosclerosis is a multifactorial inflammatory pathology that involves metabolic processes. Improvements in therapy have drastically reduced the prognosis of cardiovascular disease. Nevertheless, a significant residual risk is still relevant, and is related to unmet therapeutic targets. Endothelial dysfunction and lipid infiltration are the primary causes of atherosclerotic plaque progression. In this contest, mitochondrial dysfunction can affect arterial wall cells, in particular macrophages, smooth muscle cells, lymphocytes, and endothelial cells, causing an increase in reactive oxygen species (ROS), leading to oxidative stress, chronic inflammation, and intracellular lipid deposition. The detection and characterization of mitochondrial DNA (mtDNA) is crucial for assessing mitochondrial defects and should be considered the goal for new future therapeutic interventions. In this review, we will focus on a new idea, based on the analysis of data from many research groups, namely the link between mitochondrial impairment and endothelial dysfunction and, in particular, its effect on atherosclerosis and aging. Therefore, we discuss known and novel mitochondria-targeting therapies in the contest of atherosclerosis.

## 1. Introduction

Despite tremendous progresses in therapy in the last few years, cardiovascular diseases are still the leading cause of death worldwide [1,2]. To date, 18 million people die each year due to cardiovascular diseases [1,3,4,5,6,7]. These numbers are expected to increase, reaching 24 million yearly deaths worldwide from cardiovascular-related disease by 2030, with an average of over 66,000 per day, and a total global cost rising to over 1 trillion USD [1,3,4,5,6,7]. After a decline in mortality in recent decades, the numbers are rising again, reversing years of progress in both ischemic heart and cerebrovascular disease [1,4]. Thus, cardiovascular prevention is essential to make a 25% reduction in premature mortality from non-communicable diseases more realistic. There is an urgent need for plans and policies to reduce the burden of cardiovascular diseases which are a health challenge, starting from the territory and the role of cities in promoting health. There is also a need for improving our understanding of the basic mechanisms underlying cardiovascular diseases to design tailored therapy.

The atherosclerotic process is very often responsible for several cardiovascular and cerebrovascular diseases [8]. It is now well accepted and documented that atherosclerosis starts from endothelial dysfunction and lipid deposition, which progresses through macrophage infiltration, smooth muscle cell migration, and blood borne material deposition and becomes clinically relevant due to complications, eventually leading to local intravascular thrombus formation [8,9]. Modified lipoproteins, mainly oxidized low-density lipoproteins (oxLDL), are considered the major contributors to the genesis, progression, and immunological response occurring during the atherosclerotic process [10,11]. The first report linking mitochondria to atherosclerosis is from 1970 [12]. However, only in the last few years has increasing evidence really underlined the key role of mitochondrial dynamics in the pathogenesis of atherosclerosis [13]. Vascular cells, such as endothelial and smooth muscle cells, due to their metabolic functions and their barrier role are the main targets of mitochondrial dysfunction. In the atherosclerosis process, dysfunctional mitochondria might cause alterations in cellular metabolism and respiration resulting in the excessive production of reactive oxygen species (ROS), leading to oxidative stress [13]. While low levels of ROS exert important signaling functions [14,15], elevated ROS production induces the damage of cellular structures, alters DNA, proteins, and other molecules [16]. These conditions can become chronic, thereby favoring atherosclerosis progression and destabilization [17]. It is important to point out that mitochondrial dysfunction can be inherited and acquired [18]. Primary mitochondrial disease (PMD) is clinically diagnosed and confirmed by pathogenic mitochondrial DNA (mtDNA) or nuclear DNA (nDNA) mutations [18]. However, there are some disorders with a ‘mitochondrial’ phenotype without identifiable mtDNA or nDNA mutations, or with variants of unknown clinical significance [18]. Conversely, secondary mitochondrial dysfunction (SMD) can be caused by genes encoding neither function nor production of oxphos proteins, and associated with many hereditary non-mitochondrial diseases [18]. This SMD can also be related to nongenetic causes, such as environmental factors and ageing. The strong relationship between ageing, atherosclerosis, and cardiovascular diseases is well established [19], and a correlation between mitochondrial dysfunction–ageing, and vice-versa, cannot be ruled out [20]. In this review, we will particularly focus on the relationship between mitochondrial function and atherosclerosis, starting from its molecular basis to clinical perspectives and therapeutic options.

## 2. Mechanism of Mitochondrial Dysfunction and Molecular Implication for the Cardiovascular System

Mitochondria represent the engine of human cells generating ATP, and their function is regulated via mitophagy. However, mitochondria are also involved in the production of metabolites for protein assembling and signal transduction. Their metabolism is influenced by Ca^2+^ levels: low calcium concentrations can lead to mitochondrial dysfunction, while high levels of calcium can increase mitochondrial permeability [21].

As compared to other organs, the heart needs a high amount of energy; this is produced by oxidative metabolism in the mitochondria, which are one-third of the total volume of cardiomyocytes [22].

Mitochondrial dysfunctions are strongly related to cardiovascular diseases, in particular ischemic heart disease and atherosclerosis, cardiomyopathy, and hypertension [23]. This altered mitochondrial function results in reduced ATP, impairment of mitochondrial regulatory roles, reactive oxygen species (ROS) production and related signaling, cell growth and apoptosis, impaired mitochondrial electron transport chain activity, and an inflammatory response [24,25,26,27].

Mitochondrial fission and fusion impairments have been related to the development of cardiovascular diseases [28]. Novel mitochondrial biomarkers have been proposed to identify patients that can benefit from therapies specifically targeting mitochondria [29].

Mitochondrial homeostasis plays a crucial role in cells with high energy consumption, and in the pathological development of ischemic heart disease [30]. In particular, variation in the level of Ca^2+^ in mitochondria could lead to altered contraction coupling [31] and increased ROS levels during ischemic myocardial reperfusion, thereby inducing mitochondrial membrane damage, abnormal ATP synthesis, increased levels of Ca^2+^, and mPTP opening [32].

Thus, during ischemia, the mitochondrial succinate significantly increases in injured tissue [33,34], while during reperfusion, the accumulated metabolites are oxidized by the mitochondrial respiratory chain enzyme SDH (succinate dehydrogenase). This drives reactive oxygen species (ROS) production by reverse electron transport (RET) at mitochondrial complex I, stimulating mitochondrial permeability transition pore opening, and cell death associated with ischemia/reperfusion damage [35,36,37,38]. Sazanova et al. have demonstrated that specific mutations in the coding region of the mitochondrial genome, in particular m.14459G>A and m.5178C>A, are risk factors for occurrence and development of cardiac angina, while the mutation m.15059G>A instead had a protective effect [39].

Atherosclerosis is a chronic inflammatory condition characterized by impaired lipid metabolism, which stimulates innate and adaptive immune responses [9]. The presence of lipids in the intimal layer of arteries significantly affects mitochondrial activity/function. There are several mechanisms related to atherosclerosis progression and plaque instability, such as mutations in mitochondrial DNA (mtDNA), reactive oxidative species (ROS), and respiratory chain alterations, which induce hypertrophy of vascular smooth myocytes [40]. Mitochondrial ROS, associated with several risk factors, such as hyperglycemia and hypercholesterolemia [41], are involved in the expansion and potential damage of atherosclerotic plaques, particularly by inducing endothelial dysfunction, promoting monocyte infiltration, and increasing vascular smooth muscle cell and endothelial cell apoptosis [42].

Accumulation of ROS in the inner mitochondrial membrane can alter the mitochondrial cholesterol transporter, steroidogenic acute regulatory protein (StAR), which inhibits the storage of cholesterol through the mitochondrial membrane [43].

Under pathophysiological conditions, the dysfunctional mitochondria can stimulate the production of large amounts of ROS from adjacent mitochondria, modifying membrane potential, a process well known as “ROS-induced ROS”. Normally, this process of ROS is balanced by endogenous antioxidants. On the other hand, when there is an oxidative stress and hyper-production of ROS, the oxidative impairment in the arterial wall is increased [44], and a progression of the atherosclerotic plaque can be activated [45].

Mutations in mtDNA, related to ROS derived from mitochondrial dysfunction, are more frequent than in nuclear DNA and can cause impaired mitochondrial respiration in smooth muscle monocytes and macrophages, thereby contributing to atherosclerotic plaque progression [41]. In vitro studies have demonstrated that the damage induced byROS on mtDNA inhibits protein synthesis and alters gene expression, suggesting that ROS might mediate vascular cell dysfunction in the setting of atherogenesis [46] associated with changes in mitochondrial genome [47].

On the other hand, recent studies have put into question the correlation between ROS and mtDNA damage, highlighting that impaired mtDNA was observed in vascular and circulating cells before the development of atherosclerotic lesions [46]. Thus, these data suggested that the primary event is associated with mtDNA damage which increases ROS generation and variations in mitochondrial dysfunction and membrane potential, followed by activation of the apoptotic mechanism. Moreover, endogenous mitochondrial damage-associated molecular patterns (mtDAMPs) trigger sterile inflammation through various signaling pathways, including toll-like receptors (TLRs), nuclear factor kappa beta (NF-κβ), and the NOD-, LRR-, and pyrin domain-containing protein 3 (NLRP3) inflammasome [48]. In addition, damaged mtDNA can be seen as an endogenous damage-associated molecular pattern (DAMP), activating the inflammatory response [49].

Oxidized LDL (oxLDL) is involved in apoptosis through calcium-dependent mitochondrial pathways, mediated by cytochrome C and apoptosis-inducing factors [50]. In addition, oxLDL induces mitochondrial DNA (mtDNA) damage in atherosclerotic plaques, resulting in reduced aerobic respiration in the cap and core regions, thus decreasing the expression of complex I in vascular smooth muscle cells [45], which promotes the development of the necrotic core and the thickening of the fibrous cap in atherosclerotic plaques. Thus, oxLDL inhibits mitochondrial enzymes and modifies mitochondrial membrane potential causing production of MitoROS associated with endothelial apoptosis and atherosclerotic plaque progression [51,52]. Notably, damage to mitochondria of inflammatory cells, as macrophages or monocytes, induces plaque progression and complication, such as erosion or rupture [13].

## 3. Mitochondrial Dysfunction and Endothelial Dysfunction

Endothelial cells represent the barrier for the molecular transport between blood and tissue, regulating the exchange of nutrients [53] and the vascular tone by production of nitric oxide (NO) and contractile factors. Nevertheless, in addition to its function as a barrier, the endothelium modulates inflammation, lipid metabolism, vascular tone, and hemostasis, and prevents penetration of inflammatory cells into the bloodstream. It also secretes adhesion molecules and cytokines involved in the inflammatory cascade [54,55].

Endothelial cells’ dysfunction is characterized by ROS production, in particular from mitochondria, infiltrations of LDL, and subsequent oxidation to oxLDL [56], which might induce endothelial impairment. This enhances the expression of adhesion molecules ICAM-1, VCAM-1, and P-selectins, resulting in smooth muscle cell growth and activation of inflammatory cells, such as monocytes and macrophages, thereby causing a pro-inflammatory burden by releasing cytokines and increasing circulating immune cells’ adhesion to the endothelium [57].

Damaged endothelium produces growth factors, which activates smooth muscle cells (SMCs) in the vascular bed and the phagocytosis of lipids, with subsequent evolution into the fibrous cap [58].

Mitochondria are important regulators of apoptosis and NO production in vascular endothelial cells, stimulating the cellular response to stress [59]. At the same time, the amounts of mitochondria in the endothelium are not abundant, but despite that, mitochondria not only affect energy supply, but also regulate blood oxygen levels and mediate NO-mediated vasodilation [60,61]. The decreased NO production in atherosclerotic plaques is mainly due to endothelial NO synthase (eNOS) degradation, induced by ROS-mediated oxidative stress. In addition, eNOS decoupling dysfunction produces additional ROS, which interfere with endothelial function, inducing the progression of atherosclerosis [62].

Recent studies have highlighted that the transcription factor NF-E2 p45-related factor 2 (Nrf2; gene name *NFE2L2*) regulates some mitochondrial functions [63] via the mtROS pathway, such as antioxidant activity, autophagy, and metabolism [64]. Nrf2 has a crucial role in the maintenance of cellular redox homeostasis by regulating biosynthesis, utilization, and regeneration of glutathione, thioredoxin, and NADPH, and controlling the production of ROS by mitochondria and NADPH oxidase. Moreover, Nrf2 activation inhibits Drp1-mediated mitochondrial fission, improving endothelial dysfunction [65].

Incubation of endothelial cells with oxLDL leads to an increase in the activity of mitochondrial complex I and oxidative stress [66], thereby stimulating the transcription and expression of superoxide dismutase 2 (SOD2) in macrophages [67].

Activation of mitophagy, a lysosome-mediated selective mitochondria degradation [68], is one of the characteristic signs of mitochondrial dysfunction and damage and leads to membrane potential collapse, increased ROS production, and decreased ATP levels, oxidative stress, and apoptosis [69]. The balance between mitophagy and mitochondrial neogenesis is crucial for correct homeostasis in the human body [70]. Mitochondrial biogenesis is an important process for maintaining energy control and protecting endothelial cells’ survival in critical pathological scenarios [71,72]. Mitochondrial dysfunction might represent the first real step of atherosclerosis by determining the endothelial impairment, which is the starting point of the atherosclerotic process [21].

The endothelial dysfunction, in particular decreased vasodilatation, is also typical of MELAS (Mitochondrial myopathy, encephalopathy, lactic acidosis, and stroke-like episodes) patients [73].

These patients can have lower levels of l-arginine, which regulates the endothelial-dependent vascular relaxation [74], which was significantly lowered in both acute and interictal phases of MELAS as compared to control subjects [75].

A schematic view of the atherosclerotic mitochondria-related process is shown in Figure 1.

## 4. Clinical Perspectives and Therapeutic Implications: Something New?

In the last twenty years, several efforts have been made to evaluate pharmacological strategies to modulate mitochondrial function. Many preclinical and clinical studies have been designed based on the available literature, which support the notion that mitochondria might be crucial for the treatment of cardiovascular diseases. However, no specific studies are currently available on atherosclerosis. Starting from the concept that during cardiovascular stress mitochondrial oxidative function is enhanced, a primary therapeutic approach might be based on the reduction of this activation [23,76]. The first report targeting mitochondrial function with antioxidants (i.e., MitoQ) has reduced doxorubicin-induced cardiomyopathy in an animal model [77], and increased endothelial performance in the elderly [78]. Upcoming clinical trials (NCT03586414) will reveal the effect of MitoQ supplementation as a therapeutic modality for improving health outcomes in cardiac patients [79]. An alternative strategy to recover mitochondrial function is nicotinamide adenine dinucleotide (NAD+) supplementation [79]. It has been reported that both nicotinamide riboside and nicotinamide mononucleotide may improve cardiac function [80] and reduce stroke occurrence in several preclinical models [81]. Potential benefits of NAD+ supplementation have also been reported in heart failure patients [82]. Other general approaches might have the potential to preserve mitochondrial activity. First, it has been reported that resveratrol and curcumin may improve mitochondrial function in experimental models of heart diseases [83]. Moreover, some lifestyle interventions, such as caloric restriction and exercise, contribute to rescuing mitochondrial dysfunction [79,84]. A more specific modulation with elamipretide is under clinical investigation [85]. This is a cell-permeable small peptide SS-31 which improves homeostatic mitochondrial activities and cardiac function [85]. Despite the failure in limiting restenosis in the clinical contexts of coronary and renal angioplasty [86,87], promising results have been reported in an experimental model of heart failure [88], and it is actually under evaluation in clinical trials enrolling patients with heart failure [89].

Other specific agents targeting specific mitochondrial activities (such as mitophagy and/or autophagy, mitochondrial calcium uniporter protein, Ca^2+^-responsive enzymes, damage-associated molecular patterns, miRNA-targeting strategies) might have potential as direct or indirect therapeutical modulators of mitochondrial functions [85]. However, some of these new compounds have multiple targets, and thus it becomes difficult to establish the appropriate dose to specifically target mitochondrial dynamics [85]. Moreover, based on the available data, modulation of mitochondrial dynamics might be beneficial only in the short term, since detrimental cardiac effects may occur in the long term [85]. Thus, modulators of mitochondrial dynamics have not yet been tested on patients affected by cardiovascular diseases.

A new therapeutic perspective is characterized by neutralization of mitochondria by ROS, targeting ROS scavengers to the site of action, or by induction of mitochondrial biogenesis. Recently, new mitochondria targeted-antioxidants, preferentially delivered to the mitochondrial compartment to scavenge mitochondria-derived ROS, have been proposed to increase antioxidant levels specifically in mitochondria [90,91]. In particular, Karnewar et al. [92] have demonstrated that an an alkyl TPP + -tagged esculetin (mitochondria-targeted esculetin or Mito-Esc) reduces atherosclerosis by delaying vascular senescence and pro-inflammatory processes, and by increasing mitochondrial biogenesis. Furthermore, in humans, it has been demonstrated that pioglitazone, an anti-diabetic drug, stimulates mitochondrial biogenesis in the subcutaneous fat of patients with diabetes mellitus, a strong risk factor for atherosclerotic disease, contributing to the lipid-lowering effect of this drug [93].

Recently, Dorighello et al. [94] showed that reducing the endogenous production of mitochondria-related oxidants can counteract atherosclerosis progression, inducing a mild mitochondrial uncoupling and thus decreasing oxidant generation, resulting in a decrease in development within the hypercholesterolemic LDLr-/- model. Additionally, treatment with low doses of DNP (2,4-dinitrophenol) did not alter classical risk factors for atherosclerosis in the LDLr-/- mice, reinforcing that mitochondrial ROS in the arterial wall have a crucial pro-atherogenic role.

## 5. Current Gaps and Future Research

In view of the multiple mechanisms described above, the development of a drug that has the pharmacodynamic and pharmacokinetic characteristics that allow it to modulate mitochondrial function, with proven clinical efficacy, appears still far away.

The current knowledge of the mitochondrial mechanisms underlying cardiovascular diseases is still limited and fragmented; therefore, basic and clinical research have the primary goal to better define the pathways involved, in order to identify a more precise therapeutic target.

However, several issues should be taken into account. The first problem is mitochondrion specificity. Previously published research on antioxidants clearly indicates that in the absence of specificity, these agents do not act on mitochondrial activity only and inhibit reactive oxygen species (ROS) generated from multiple sources, with controversial clinical benefits. Improved pharmacological strategies now offer the possibility of targeting antioxidants to the mitochondria using cationic molecules which have the ability to accumulate spontaneously in the mitochondrial matrix [95,96]. However, the dysfunctional mitochondrion could present transmembrane potential abnormalities, thus altering the mitochondrial capacity to accumulate the drug delivered by cation-based strategies [97,98,99].

The second problem is the pharmacokinetic activity and biodistribution of the potential drug. To date, a tissue-specific strategy is missing. With systemic administration, any medication inevitably interacts with mitochondria, not only of the cardiovascular system, thus limiting the bioavailability, at the doses deemed safe, of some MTAs in the sites of interest [85].

It is known that several mitochondrial proteins are necessary for embryonic development and/or adult survival, especially for their bioenergetic functions. An example of this essential role is somatic cytochrome c (CYCS), which plays the role of electronic shuttle of the respiratory chain [100] and whose absence leads to in uterus death [101]. Unfortunately, many mitochondrial proteins exist in multiple isoforms and with multiple functions [102]. Genetic and functional redundancy complicates the generation of suitable experimental models [85].

Similarly, it is difficult to monitor mitochondrial function in vivo. Despite the possibility to measure the carbonylation of circulating proteins or lipoproteins to monitor oxidative stress in the context of CVD [103], this technique is not tissue specific and thus makes it impossible to clearly define the source of ROS. In the last few years, technological advancements have made it possible to measure, more specifically, the carbonylation of cardiac proteins, such as myosin binding protein C, cardiac type (MYBPC). However, it is not really specific for the identification of the ROS source, and it can be performed only post-mortem [104].

Recently, energy metabolites measured by mass spectrometry have been proposed as potential markers. However, the clinical applicability remains a matter of debate [105]. More promising results are instead linked to the MitoTimer mouse, which represents the most advanced experimental model of mitochondrial dysfunction in the preclinical cardiovascular field [106,107] providing the possibility of studying the mitochondrial structure, the redox state, and the mitophagy function [106,107].

Furthermore, a series of radioactive tracers seem to represent a good potential for monitoring mitochondrial functions in the CV field [108]. In this context, pre-specified studies could open a new frontier to better evaluate the link between mitochondrial dysfunction and multiple forms of CVD.

In the last decade, several experimental evidences on animal models have pointed out a significant gender related difference in CVD development [109]. These observations are also supported by real-life epidemiological data [110,111,112]. However, the evaluation of this aspect in correlation with mitochondrial dysfunction is completely missing.

Finally, the time factor should be considered. Atherosclerosis is a time-dependent chronic disease and at the moment of its clinical relevance, the etiopathogenetic factors have already reached a degree of expression which in some cases is irreversible [8]. Atherosclerosis-mediated CVDs mainly involve older individuals with other comorbidities beyond age, such as obesity, hypertension, diabetes, and immune frailty, thus defining progression and treatment sensitivity of CVD [113,114]. To date, few experimental models reproduce this clinical condition, further limiting our knowledge on the mixed contribution of these comorbidities on mitochondrial function.

Taking together all these limitations, future research should be directed to (a) improve the pharmacodynamic and pharmacokinetic properties of therapeutic strategies, (b) study in depth the mechanisms related to mitochondrial function, (c) reevaluate the pathogenesis of CVDs, or at least those such as atherosclerosis which strongly correlate with mitochondrial dysfunction.

## 6. Conclusions

Atherosclerosis is a multifactorial disease. Multiple clinical trials and basic studies have clearly demonstrated that the management of the known risk factors only is not enough to limit the burden of this condition which underlies most cardiovascular diseases. Increasing evidence suggests an important role for mitochondria in the initial steps of this process. They regulate the inflammatory response and oxidative stress, two key steps that, once dysfunctional, might modulate initiation and progression of the atherosclerotic lesion. Thus, the modulation of mitochondrial function could delay the development of endothelial dysfunction, which represents the primum movens of the atherosclerotic process. In this context, it will be important for research, at both preclinical and clinical levels, to define the precise therapeutic interventions focusing on mitochondrial functions, considering all the aforementioned factors such as age, sex, and relevant comorbidities.

In this review, we have summarized the latest data on the relationship between mitochondrial functions and atherosclerosis, with a look in depth at endothelial dysfunction. Although mitochondria have been recognized as a new therapeutic target in different pathological contests, the current gaps in therapeutic strategies to manage mitochondrial dysfunction do not allow for the provision of a clinical benefit in the development of atherosclerosis. Both antioxidants and gene therapy are attractive approaches for the treatment of atherosclerosis. However, further ongoing and new studies are needed to identify the correct strategy for reducing the impact of mitochondrial dysfunction on the progression of atherosclerosis.

## Figures and Tables

**Figure 1 ijms-24-01086-f001:**
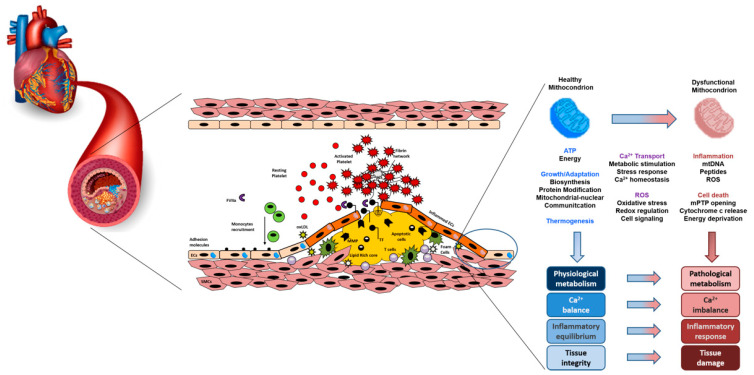
A schematic view of the interaction between mitochondria dysfunction, atherosclerosis and oxidative stress.

## Data Availability

Not applicable.

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
