# Peer review of "Mitochondrial Dysfunction: The Hidden Player in the Pathogenesis of Atherosclerosis?"

_ijms, 2023, doi:10.3390/ijms24021086_

Round 1

Reviewer 1 Report

Review on the manuscript “Mitochondrial dysfunction: the hidden side in the pathogenesis of atherosclerosis?” by Giovanni Ciccarelli, Stefano Conte, Giovanni Cimmino, and Antonio Giordano (Manuscript ID: ijms-2068130).

First of all, I should note that presented review is devoted to a very relevant topic in a field - the problem of the development of atherosclerosis and the direct participation of mitochondria in this process.

The presented manuscript is consistent with the mission of the journal and executed in accordance with its requirements.

The narrative language is clear and accessible, and should not cause misunderstanding on the part of the reader.

However, there are some issues that must be fixed:

1.      The authors paid enough attention to highlight the role of mitochondria as the main components that regulate the inflammatory response and oxidative stress, the processes that are involved in the initiation and progression of atherosclerotic lesions.

But this phenomenon is quite well known and have been stressed out in many original articles and reviews (e.g. reviews by Chen Y et al. (doi: 10.1089/dna.2022.0249) or published in Int J Mol Sci. 2022 by Poznyak AV et al. (doi: 10.3390/ijms23062951)). Unlike the sources I have cited, after reading the presented text of the manuscript, it remains unclear, so what exactly is the originality of the authors' point of view on this problem?

2.      The authors note that despite a fairly large number of studies indicating the important role of mitochondria in the development of atherosclerosis, no specific therapeutic approaches to modulate the early stage of atherosclerosis with the participation of mitochondria have been developed to date. The authors suggest that an approach based on the modulation of mitochondrial function "may be an attractive therapy for several cardiovascular diseases, including atherosclerosis."

These statements are too vague, and therefore they do not carry any semantic load. Authors need to provide their readers with their original ideas or approaches that could make it possible to implement this idea, at least in theory, but nothing of the kind is considered. (e.g. see review by Stamerra CA et al. (doi: 10.1155/2022/9530007); or review by Luan Y et al. (doi: 10.3389/fcvm.2021.770574)).

Because the review by Ciccarelli et al. undoubtedly refers to an interesting topic, and if it were presented in a different way, it could attract the attention of not only narrow specialists, but also be of interest to researchers of a fairly wide range.

I suggest that the authors shall make a major revision of the manuscript, rewriting the text in the form of a dispute with already published articles, while emphasizing the originality and novelty of their own ideas and inviting the reader to the discussion.

Author Response

Reviewer #1

First of all, I should note that presented review is devoted to a very relevant topic in a field - the problem of the development of atherosclerosis and the direct participation of mitochondria in this process. The presented manuscript is consistent with the mission of the journal and executed in accordance with its requirements.The narrative language is clear and accessible, and should not cause misunderstanding on the part of the reader.

However, there are some issues that must be fixed:

  1. The authors paid enough attention to highlight the role of mitochondria as the main components that regulate the inflammatory response and oxidative stress, the processes that are involved in the initiation and progression of atherosclerotic lesions.

But this phenomenon is quite well known and have been stressed out in many original articles and reviews (e.g. reviews by Chen Y et al. (doi: 10.1089/dna.2022.0249) or published in Int J Mol Sci. 2022 by Poznyak AV et al. (doi: 10.3390/ijms23062951)). Unlike the sources I have cited, after reading the presented text of the manuscript, it remains unclear, so what exactly is the originality of the authors' point of view on this problem?

Authors’s response: we agree with the Reviewer e we have added a new paragraph and edited the previous version of the manuscript in order to better clarify the originality of this review.

Changes made: changes in bold in paragraph 4 and new paragraph 5

  1. The authors note that despite a fairly large number of studies indicating the important role of mitochondria in the development of atherosclerosis, no specific therapeutic approaches to modulate the early stage of atherosclerosis with the participation of mitochondria have been developed to date. The authors suggest that an approach based on the modulation of mitochondrial function "may be an attractive therapy for several cardiovascular diseases, including atherosclerosis."

These statements are too vague, and therefore they do not carry any semantic load. Authors need to provide their readers with their original ideas or approaches that could make it possible to implement this idea, at least in theory, but nothing of the kind is considered. (e.g. see review by Stamerra CA et al. (doi: 10.1155/2022/9530007); or review by Luan Y et al. (doi: 10.3389/fcvm.2021.770574)).

Because the review by Ciccarelli et al. undoubtedly refers to an interesting topic, and if it were presented in a different way, it could attract the attention of not only narrow specialists, but also be of interest to researchers of a fairly wide range.

I suggest that the authors shall make a major revision of the manuscript, rewriting the text in the form of a dispute with already published articles, while emphasizing the originality and novelty of their own ideas and inviting the reader to the discussion.

Authors’s response: Again, thank the Reviewer for his suggestions. We provide an extensive revision of the manuscript and we think to have better specified the originality and the novelty of this paper.

Changes made: extensive review of the full manuscript as compared to the previous version.

Reviewer 2 Report

The review is based on a very interesting concept, i.e. clarifying the pathogenetic basis of atherosclerotic damage related to the mitochondrial dysfunction, in order to have more specific therapeutic targets.

However, the premise is not satisfied by the body of the review, which appears based on a elencation of examples of different mitochondrial involvement reported by previous studies, without a real insight into the pathogenetic mechanisms involved in the various aspects of atherosclerotic pathology.

The reference to primary mitochondrial pathologies, which instead could have served as an explanatory basis for some aspects of ischemic damage, is also not discussed in depth. There is no mention of the involvement of the endothelial damage in the pathogenesis of stroke-like episodes in the MELAS syndrome (Mitochondrial encephalomyopathy, lactic acidosis, and stroke-like episodes).

In adjunction, the paper carries stylistic and linguistic inaccuracies which, overall, make the text difficult to read:

Lines 65-73, English language need to be improved

Line 94-95 is a repetition

Line 116, “conditions” is not correct, please amend

Lines 117-120: no clear sentence, please rephrase

Line 177: change “poor” with “poorly”

Author Response

Reviewer #2

The review is based on a very interesting concept, i.e. clarifying the pathogenetic basis of atherosclerotic damage related to the mitochondrial dysfunction, in order to have more specific therapeutic targets.

However, the premise is not satisfied by the body of the review, which appears based on a elencation of examples of different mitochondrial involvement reported by previous studies, without a real insight into the pathogenetic mechanisms involved in the various aspects of atherosclerotic pathology.

The reference to primary mitochondrial pathologies, which instead could have served as an explanatory basis for some aspects of ischemic damage, is also not discussed in depth. There is no mention of the involvement of the endothelial damage in the pathogenesis of stroke-like episodes in the MELAS syndrome (Mitochondrial encephalomyopathy, lactic acidosis, and stroke-like episodes).

Authors’s response: thank the Reviewer for his suggestion.

Changes made: changes in bold page 5 line 218-223

In adjunction, the paper carries stylistic and linguistic inaccuracies which, overall, make the text difficult to read:

Lines 65-73, English language need to be improved

Line 94-95 is a repetition

Line 116, “conditions” is not correct, please amend

Lines 117-120: no clear sentence, please rephrase

Line 177: change “poor” with “poorly”

Authors’s response: we agree with the Reviewer e we have provided a full extensive English revision of the manuscript by Andrea Morrione that is now added as a new Author.

Changes made: the entire manuscript has been revised.

Reviewer 3 Report

The manuscript entitled “Mitochondrial dysfunction: the hidden side in the pathogenesis of atherosclerosis?” by G. Ciccarelli et al., was purposed as Review in a section of “Molecular Pathology, Diagnostics, and Therapeutics” in which the authors set out to investigate the potential link between mitochondrial and endothelial dysfunction and their effects on atherosclerosis and aging. Unfortunately, I found the quality of the Review not satisfactory at different levels. In general, the text presents many problems with written English. Not just because of the several typos, but, in particular, the way in which the sentences are organized require a thorough revision. From the title the aim of the review is to explain how alteration in term of mitochondrial function are involved in atherosclerosis. In the body of the manuscript, instead, is reported mechanism of mitochondrial dysfunction in the cardiovascular system, without specifically and newly explanations of its illness involvement. The novelty of the review is lacking and, although the topic may be interesting, I do not understand the contribution in term of scientific article. It should be better stressed the role of mitochondrial dysfunction in the pathogenesis of atherosclerosis and their clinical perspectives and therapeutic implications. Discussion is too short.  In the discussion section, it is considered helpful to highlight current problems and a future perspective on this topic. The figure 1 is not so clear with respect to the legend. 

Author Response

Reviewer #3

The manuscript entitled “Mitochondrial dysfunction: the hidden side in the pathogenesis of atherosclerosis?” by G. Ciccarelli et al., was purposed as Review in a section of “Molecular Pathology, Diagnostics, and Therapeutics” in which the authors set out to investigate the potential link between mitochondrial and endothelial dysfunction and their effects on atherosclerosis and aging. Unfortunately, I found the quality of the Review not satisfactory at different levels. In general, the text presents many problems with written English. Not just because of the several typos, but, in particular, the way in which the sentences are organized require a thorough revision. From the title the aim of the review is to explain how alteration in term of mitochondrial function are involved in atherosclerosis. In the body of the manuscript, instead, is reported mechanism of mitochondrial dysfunction in the cardiovascular system, without specifically and newly explanations of its illness involvement. The novelty of the review is lacking and, although the topic may be interesting, I do not understand the contribution in term of scientific article. It should be better stressed the role of mitochondrial dysfunction in the pathogenesis of atherosclerosis and their clinical perspectives and therapeutic implications. Discussion is too short.  In the discussion section, it is considered helpful to highlight current problems and a future perspective on this topic. The figure 1 is not so clear with respect to the legend

Authors’s response: we agree with the Reviewer e we have provided a full extensive English revision of the manuscript by Andrea Morrione that is now added as a new Author.

Therefore, we rewrite some paragraph in order to better clarified the involvement of mitochondria in atherosclerosis, and the clinical perspectives and therapeutic implications. Also, a new paragraph has been added to discuss about current gaps and future research. Figure 1 has been full edited.

Changes made:

The entire manuscript has been revised in order to improve written English.

Paragraph 2, line 134-140
Paragraph 4, line 258-261 and 268-287

New paragraph 5

Paragraph 6, line 362-365 and 368-376

A new version of Figure 1 has been added to the manuscript.

Reviewer 4 Report

This is a review focalizing on the relationship between mitochondrial function and atherosclerosis. The authors start from molecular basis to clinical and therapeutic perspectives.

Although very concise, the review highlights important aspects of the molecular mechanisms operating at the cross-talk between mitochondria and cellular dysfunction with a special focus on endothelium and cardiovascular networks.

Only a minor revision is needed:

-          In line 25, 53, 80, 83 and 118 it should be correct the text regarding “mitochondria”

-          In line 118 it should be deleted “or”

-          The sentence reported in line 152-155 should be improved in term of clarity

-          The Figure 1 should be edited to provide a clearer scheme of the specific issue

Author Response

Reviewer #4

This is a review focalizing on the relationship between mitochondrial function and atherosclerosis. The authors start from molecular basis to clinical and therapeutic perspectives.

Although very concise, the review highlights important aspects of the molecular mechanisms operating at the cross-talk between mitochondria and cellular dysfunction with a special focus on endothelium and cardiovascular networks.

Only a minor revision is needed:

-          In line 25, 53, 80, 83 and 118 it should be correct the text regarding “mitochondria”

-          In line 118 it should be deleted “or”

-          The sentence reported in line 152-155 should be improved in term of clarity

-          The Figure 1 should be edited to provide a clearer scheme of the specific issue

Authors’s response: we agree with the Reviewer e we have provided a full extensive English revision of the manuscript by Andrea Morrione that is now added as a new Author.

Changes made: the entire manuscript has been revised. A new version of Figure 1 has been added to the manuscript.

Round 2

Reviewer 1 Report

Please see file

Author Response

I very carefully looked at the revised manuscript.

Despite an effort the authors put into revised version, they did not achieve the goals that I

asked them to make at the first time.

  1. First of all, I simply have to express my bewilderment at the approach taken by the

authors in the revised version of the manuscript.

Instead of making a local substitution and/or addition of an article, word, or phrase in a

sentence, the authors cross out an entire paragraph, and below they insert an almost exact copy

of the same text with minor changes.

Examples are given below, the original text is marked in blue, the altered text is in black,

the words that have been replaced or rearranged are highlighted in red. In 1.Introduction:

The first report linking mitocondria to atherosclerosis is dated 1970 [12]. However, only in the last

few years increasing evidence have really underlined the key role of mitochondrial dynamics in the

pathogenesis of atherosclerosis.

The first report linking mitochondria to atherosclerosis is from 1970 [12]. However, only in the last

few years increasing evidence have really underlined the key role of mitochondrial dynamics in the

pathogenesis of atherosclerosis [13].

, elevated ROS production induces damage of cellular structures, alters DNA, proteins, and other

molecules [16]. These conditions become chronic favoring atherosclerosis progression and

destabilization [17]. It is important to point out that mitochondrial dysfunction may be inherited and

acquired [18]. Primary mitochondrial disease (PMD) is diagnosed clinically confirmed by a pathogenic

mitochondrial DNA (mtDNA) or nuclear DNA (nDNA) mutation [18]. However, there are some disorders

with a ‘mitochondrial’ phenotype without an identifiable mtDNA or nDNA mutation or with a variant of

unknown clinical significance [18]. Conversely, secondary mitochondrial dysfunction (SMD) may be

caused by genes encoding neither function nor production of the oxphos proteins and may be

associated to many hereditary non-mitochondrial diseases [18]. This SMD may also be related to

nongenetic causes such as environmental factors and ageing. It is well known that a strong relationship

between ageing, atherosclerosis and cardiovascular diseases exists [19]. Moreover, the correlation

mitochondrial dysfunction-ageing and vice-versa cannot be rule out [20]. In this review we will focus

mainly on the relationship between mitochondrial function and atherosclerosis, starting from molecular

basis to clinical perspectives and any possible therapeutical option available.

, elevated ROS production induces damage of cellular structures, alters DNA, proteins, and other

molecules [16]. These conditions can become chronic, thereby favoring atherosclerosis progression and

destabilization [17]. It is important to point out that mitochondrial dysfunction can be inherited and

acquired [18]. Primary mitochondrial disease (PMD) is clinically diagnosed and confirmed by pathogenic

mitochondrial DNA (mtDNA) or nuclear DNA (nDNA) mutations [18]. However, there are some disorders

with a ‘mitochondrial’ phenotype without identifiable mtDNA or nDNA mutations or with variants of

unknown clinical significance [18]. Conversely, secondary mitochondrial dysfunction (SMD) can be

caused by genes encoding neither function nor production of 9 oxphos proteins and associated to many

hereditary non-mitochondrial diseases [18]. This SMD can be also related to nongenetic causes such as

environmental factors and ageing. The strong relationship between ageing, atherosclerosis and

cardiovascular diseases is well established [19], and a correlation between mitochondrial dysfunction-

ageing and vice-versa cannot be rule out [20]. In this review we will particularly focus on the relationship

between mitochondrial function and atherosclerosis, starting from its molecular basis to clinical

perspectives and therapeutical options.

A similar trick can be seen in section 2. Mechanism of mitochondrial dysfunction and

molecular implication for the cardiovascular system: "As compared to other organs... and

further in the text", and in section 3. Mitochondrial dysfunction and endothelial dysfunction:

"At the same time... and so on".

By the way:

“... a correlation between mitochondrial dysfunction-ageing and vice-versa cannot be rule out.”

Past tense should be “cannot be ruled out.”

Thus, the appearance of a great deal of work done on the text is achieved, which, in fact,

has not actually been carried out. I am very upset by this fact, and personally, I strongly believe

that such an approach when working with text could not be acceptable.

Author’s response: We really want to thank Reviewer#1 for the big effort done in reviewing our paper.

As requested by others Reviewers, we have provided a full extensive English revision of the manuscript.

The examples described by Reviewer#1 are only related to the aforementioned editing and, for this reason, they appear as a copy of the previous manuscript with minor changes.

As mentioned in the previous revision, the major corrections were highlighted in bold in paragraph 2-3 and 4.

We apologize for the misunderstanding.

  1. In the end of the Introduction the authors make a statement “In this review we will

particularly focus on the relationship between mitochondrial function and

atherosclerosis, starting from its molecular basis to clinical perspectives and

therapeutical options.”

Summarizing the major concepts – is not enough in nowadays...”bring something new

and delicious on the table”.

As I asked in my first review, “Authors need to provide their readers with their original ideas or

approaches that could make it possible to implement this idea, at least in theory, but nothing of the kind is

considered.”, I have to stress it out again: Authors need to say something like this:

“In this review, we will focus on a new idea, based on the analysis of data from many

research groups, but never previously considered, namely: ... Here, present your idea, its

novelty, and originality.”

Author’s response: We have better clarified in the paragraph introduction and conclusions the aim of this review.

Changes made: page 1, line 30-33; page 8, line 359-367

  1. Here I must express my respect to the authors for the new section 5. Current gaps and

future research. This is a very smart and correct move.

In this section, the authors should have just presented their new vision of the problem, and

logically argued their personal idea, or the concept they developed.

However, the manuscript in its current state cannot be recommended for publication, as it

still requires further work

Author’s response: We appreciate the comment of the Reviewer#1

Reviewer 2 Report

Manuscript has been significantly improved, not only in the language and syle but also in the content. There are more in-depth explanations of the etiopatogenetic mechanisms under the mithocondrial involvement in atherosclerosis. Also the new paragraph: "Current gaps and future research", appears appropriate and comprehensive.

There are minor spell mistakes throughout the manuscript, which require revision.

Author Response

We'd like to thank Reviewer#2 for the revision.

We provided a new accurate editing of the manuscript

Reviewer 3 Report

To the Authors:

General comments:

The authors successfully revised the manuscript according to the comments.

The illustration is not very nice for the readers, so it needs to be completely redone. 

Author Response

Author's response: We'd like to thank Reviewer#3 for the revision.

We have provided a new image to better clarify the role of mitochondria in the patogenesis of atherosclerosis.

changes made: (figure 1)

Round 3

Reviewer 1 Report

I appreciate the work done by the authors, and I believe that this article, in this form, can be published in IJMS.